# Water Saturated with Pressurized CO_2_ as a Tool to Create Various 3D Morphologies of Composites Based on Chitosan and Copper Nanoparticles

**DOI:** 10.3390/molecules27217261

**Published:** 2022-10-26

**Authors:** Katerina S. Stamer, Marina A. Pigaleva, Anastasiya A. Pestrikova, Alexander Y. Nikolaev, Alexander V. Naumkin, Sergei S. Abramchuk, Vera S. Sadykova, Anastasia E. Kuvarina, Valeriya N. Talanova, Marat O. Gallyamov

**Affiliations:** 1Faculty of Physics, Lomonosov Moscow State University, Leninskie Gory 1-2, 119991 Moscow, Russia; 2A. N. Nesmeyanov Institute of Organoelement Compounds, Russian Academy of Sciences, Vavilova 28, 119334 Moscow, Russia; 3FSBI Gause Institute of New Antibiotics, Bol’shaya Pirogovskaya 11, 119021 Moscow, Russia

**Keywords:** porous hydrogels, capsules, reusable catalysts, nitrobenzene reduction, antimicrobial activity

## Abstract

Methods for creating various 3D morphologies of composites based on chitosan and copper nanoparticles stabilized by it in carbonic acid solutions formed under high pressure of saturating CO_2_ were developed. This work includes a comprehensive analysis of the regularities of copper nanoparticles stabilization and reduction with chitosan, studied by IR and UV-vis spectroscopies, XPS, TEM and rheology. Chitosan can partially reduce Cu^2+^ ions in aqueous solutions to small-sized, spherical copper nanoparticles with a low degree of polydispersity; the process is accompanied by the formation of an elastic polymer hydrogel. The resulting composites demonstrate antimicrobial activity against both fungi and bacteria. Exposing the hydrogels to the mixture of He or H_2_ gases and CO_2_ fluid under high pressure makes it possible to increase the porosity of hydrogels significantly, as well as decrease their pore size. Composite capsules show sufficient resistance to various conditions and reusable catalytic activity in the reduction of nitrobenzene to aniline reaction. The relative simplicity of the proposed method and at the same time its profound advantages (such as environmental friendliness, extra purity) indicate an interesting role of this study for various applications of materials based on chitosan and metals.

## 1. Introduction

In recent decades, one of the promising areas of research is the creation and study of composites based on metal–organic compounds. Such materials have a wide range of possible applications, namely, the storage and separation of gases and vapors, drug delivery, and chemical reactions with catalysts [1]. These composites can be obtained by combining metal ions with organic compounds and further formation of a coordination network, which makes the resulting materials stable and helps to overcome the problem of nanoparticles aggregation. Moreover, for example, to ensure a pronounced antimicrobial activity in the microbial water treatment applications, a sufficiently large amount of nanoparticles is required [2], which in a free form distributed over a solution can themselves pollute the environment. Thus, an organic carrier for nanoparticles such as a polymer is highly desirable. Inert gases, surfactants, ionic liquids, and polymers can be used as nanoparticle stabilizers preventing their aggregation [3,4,5,6]. The method of obtaining and stabilizing metal nanoparticles with polymers is based on the binding of metal ions with macromolecules and subsequent reduction of the former. Since the resulting metal nanoparticles are compact, while they have a large specific surface area, thus the small size of the nanoparticles makes it possible to increase the functionality of the solution or substrate in which they are placed (this property is crucial for nanoelectronics and photonics [7,8]), and their surface allows them to interact more efficiently and quickly with various substances (important for catalysis problems [9]). The use of copper in ionic form for various applications has recently become increasingly popular due to the following circumstances. Among various metals, Cu is one of the most earth-abundant relatively low-cost materials, which can be used as a viable alternative to the rare and expensive noble metals. It is known that copper nanoparticles as well as Cu^2+^ ions can exhibit antimicrobial activity against test cell cultures of Gram-positive and Gram-negative microorganisms, as well as other common pathogens while traditional methods of bacteria control, such as the use of antibiotics, often become ineffective due to the development of pathogen resistance [2,10]. Moreover, copper nanoparticles are especially attractive for catalysis purposes, since they allow reactions to be carried out under “green” or sustainable conditions, limiting the use of conventional catalysts [9,11].

In this work, it was decided to use chitosan (CS), which is a derivative of chitin, to stabilize copper nanoparticles. This natural polymer is obtained by processing food industry waste, which is especially important now, when all over the world they are trying to take a more thoughtful and rational approach to both the consumption of exhaustible raw materials and waste disposal. Indeed, chitosan is a derivative of the linear polysaccharide chitin, which is a natural widely available polymer. Chitosan can be obtained by its deacetylation. The scientific interest in chitosan is determined due to its antimicrobial, wound healing and hemostatic properties, as well as pronounced biological activity, biocompatibility and non-toxicity [12,13]. At the same time, there are functional amino groups in the structure of chitosan, which makes it possible to create various composites, for example, with other polymers, as well as with metal nanoparticles. Chitosan is commonly used in biomedical applications in the forms of solutions, gels, wound dressings or patches, as human skin substitutes for treating mucosal injuries and burns, thin coatings for prostheses, and for the regeneration of bone tissues [14]. The use of chitosan as an effective adsorbent is also known. It is proven that heavy metal ions, when they enter a living organism, can be absorbed into tissues from the digestive tract, react with proteins, destroy nucleic acids, and pass from the mother’s body to the embryo. Therefore, adsorption of heavy metal ions by chitosan granules from wastewater is a promising water purification method, which can be used in leather industry and waste utilization [15,16].

There are many interesting methods for creating of various chitosan structures and forms, for example, nanofibers, made under electric field that are promising material for biomedical application due to its similarity to collagen fibers [17], flexible films [18], or multi-membrane hydrogels [19]. Moreover, chitosan solution after exposition in a base bath can form porous microspheres [20], when CO_2_ supercritical drying of such composites leads to obtaining of porous aerogels [21]. Thus, chitosan is a promising functional material.

There are many different applications of nanocomposites based on chitosan and copper. For example, they are used to minimize toxic effects and impart antifungal activity in agriculture as an alternative to chemicals of synthetic origin [22]. Composite chitosan hydrogels based on copper nanoparticles can also be used as wound dressings for medical purposes with a pronounced healing effect [23]. Moreover, chitosan-stabilized copper sulfide nanoparticles are applicable for cancer therapy [24]. Another possible application of composite chitosan hydrogels is the opportunity of using them as metal–organic framework structures for gas adsorption, gas-selective membranes, and promising catalysts [1]. Moreover, copper–chitosan composites can be used as promising catalysts, for example, in reactions of nitrophenol [25], nitroaniline [26], and oxygen [27] reduction reactions. It is known that composites based on copper and chitosan are used as antibacterial agents [28,29], moreover, with the possibility of green synthesis of such materials [30].

In practice, 1% acetic acid is most often used as a chitosan solvent due to its availability. Thus, the traditional method for creating a chitosan composite with copper nanoparticles is mixing a solution of chitosan in 1% acetic acid with a copper sulfate solution and subsequent reduction of stabilized copper ions with special reducing agents with additional thermal treatment [31]. However, in particular cases, acetic acid traces can act as haptens and stimulate an allergic reaction in the human body when used in medical applications. In addition, for a number of applications, such as heterogeneous catalysis or biomedical appliances, the composites must be additionally purified from the solvent residues [32,33]. Indeed, it is known that the most important properties of catalysts are activity, selectivity, and stability, which depend on factors such as temperature, pressure, nature, and purity of the reagents. The presence of impurities in the mixture can lead to deactivation of the catalyst due to the poisoning of active centers by the adsorption of impurities on them or to the occurrence of side reactions. These features limit the applicability of chitosan solutions in traditional acids for applications in which the purity of the compounds used is particularly important. A possible solution to the specified problem is to use a biphase H_2_O/CO_2_ system under high CO_2_ pressure as a solvent for chitosan. H_2_O/CO_2_ contains a polar aqueous medium saturated to some extent with non-polar carbon dioxide molecules dissolved in it. Carbon dioxide molecules dissolve in water, reacting with it and forming carbonic acid, which subsequently dissociates into carbonate anions and protons. It is known that at a pressure of several hundred atmospheres, the pH of the aqueous phase of the H_2_O/CO_2_ biphase system is less than 3 [34]; therefore, this system turns out to be suitable for chitosan dissolution [35]. Moreover, it was shown that in such solutions of chitosan in the presence of carbonic acid under high pressure CO_2_, it is possible to suppress the aggregation of chitosan macromolecules, which is present at the macrolevel in solutions of chitosan in traditional acids at normal pressure [36], while polysaccharide chains self-organize into nanosized aggregates consisting of only a few macromolecules. At the same time, the H_2_O/CO_2_ biphase system is a harmless medium under normal conditions (after decompression) with additional bactericidal properties that appear under pressure. The sterilizing ability of such carbonic acid solutions is caused by the presence of high-pressure CO_2_ dissolved in this medium in combination with its acidity. Under the high-pressure, CO_2_ molecules dissolved in water penetrate into bacteria and viruses dispersed in the aqueous phase and inactivate them through a number of mechanisms previously described in the literature [37]. The decompression stage is also crucial for pathogen inactivation. Another advantage of this medium is that the product formed in such a system is not contaminated with any traces of solvents, since the CO_2_ spontaneously self-eliminates after decompression, while the resulting product remains in the water. Previously, chitosan solutions in the presence of carbonic acid have been used to develop composites based on chitosan and silver, platinum, and gold nanoparticles [38,39,40], but the possibilities of composites structure and form modification were not explored. It is also worth noting that copper is a transition metal, that differs somewhat from noble ones in its properties, which is why a detailed review of the characteristics of gels based on copper and chitosan is quite useful. Moreover, in the carbonic acid solutions under the high pressure of CO_2_ chitosan aggregates differently than in conventional solutions [35].

Different applications require different 3D modifications of composites architectures. For example, for the materials with antimicrobial protection, thin films or sponges are suitable to ensure the best fit to the wound or to introduce the additional hemostatic properties in the material [41,42]. In the tasks of water purification from microbial contamination, porous sponges or capsules are needed, which can be removed from the solution after use. They should have large surface area to ensure a high concentration of deposited nanoparticles involved in purification [43]. For catalysis tasks, it is necessary to search for composites, which also can be easily removed from the reaction mixture at the end of the reaction and reused [44].

Most of the articles on copper–chitosan systems are about the sorption properties of chitosan granules or flakes [45,46,47]. Some works devoted directly to the formation of composite materials do not yet contain a complete analysis of the properties of hydrogels [31,48]. Moreover, the possibility of their architecture modification is also not considered. All this points to the need to expand the range of possible architectures of composite materials based on chitosan and copper nanoparticles so that the eventually obtained materials most effectively meet the tasks set. Moreover, in many cases, various modifications of composite materials are created by rather complex methods using special equipment, while environmentally friendly relatively economical protocols are not always possible to use.

We want to fill this gap and study copper–chitosan composite hydrogels (Cu/CS) in detail using a number of experimental techniques, as well as propose several possible 3D form modifications suitable for the applications described above. The features of stabilization and reduction of copper ions with chitosan dissolved in an ecological, self-neutralizing, and harmless carbonic acid under high CO_2_ pressure are considered.

## 2. Results and Discussion

### 2.1. Rheological Study

We found that when an aqueous solution of copper sulfate is added to a solution of chitosan in the presence of carbonic acid, a hydrogel is formed within a few seconds (see Figure 1). To study its mechanical properties, the rheology method for freshly prepared samples, based on low molecular weight chitosan, was used. It can be seen from the graphs in Figure 2a that the storage modulus exceeds the loss modulus by approximately 8 times, in other words, the distinctive feature of elastic gel (G′ > G″) remains unchanged over time (at frequencies above 100 rad/s). It is well known that the power law G′(w)~w^2^, G″(w)~w is valid for polymer melts at low frequencies and for dilute aqueous solutions of polymers over the entire frequency range. In elastic gels, both moduli show a plateau even at low frequencies [49]. However, this theory is in good agreement with the behavior of synthetic polymer gels. On the other hand, biological systems, which are ‘heterogeneous dispersions’, exhibit considerable variability in their rheological properties because of their complexity. This is why the application of classical rheological models for the description of biopolymers behavior is limited [50]. Many weak gels can be treated as three-dimensional networks where weak interactions ensure structure stability. The Gabriele, De Cindio, and D’Antona weak gel model [51] suggests that the structure of the material corresponds to the cooperative arrangement of flow units forming interacting strands. Their model introduces a coordination parameter (z), which is the number of flow units interacting with each other. During dynamic oscillatory experiments, gel strands may be considered as a combination of flow units, where z is the number of rheological units interacting with each other in a three-dimensional structure. When the loss modulus is small: G′ >> G″—the model formula is G′(w) = AFw^1/z^. It can be observed that there are two specific regions in the frequency dependence of the studied here gels: from 0.01 to approximately 60 rad/s, a nonlinear increase in G’ occurs (the behavior is consistent with the model described above), while from 60 to 200 rad/s, a plateau for G’ is observed, which is typical for the standard behavior of polymer gels. A similar behavior of the elastic moduli at high frequencies was observed earlier chitosan flake gels [52]. The network formed in the case of each gel could not be broken even at a higher angular frequency, resulting in the absence of crossover of the G′ and G′′ over the frequency range. Therefore, the gels remained viscoelastic and did not transform into viscous liquids. One can see a more detailed description and discussion on the kinetics of hydrogels’ rheological properties in the Appendix A.

After comparing the rheological characteristics of various chitosan hydrogels with metal ions, it was found that the value of the elastic modulus is similar to that for hydrogels based on chitosan and silver [38] or iron [53] and is approximately 30 times higher than the characteristics of a gel with platinum [39]. In another work [54], gels created using alginate and various metal cations, namely Ni^2+^, Co^2+^, Fe^3+^, Mn^2+^, and Cu^2+^, were considered. It was noted that the best performance was obtained for gels with iron and copper, but the elastic modulus of the former was still higher. The authors explained that alginate forms square planar complexes with two carboxyl groups, and two water molecules binding to the divalent cations, and the interaction energy is maximum with the Cu^2+^ ions among the studied divalent cations, while trivalent cations have a larger coordination number and bind with three carboxyl groups and three water molecules resulting in a more compact bonding network compared to the divalent cations. The foregoing probably explains the high mechanical characteristics of the composite obtained and the rapid formation of the gel in this work.

### 2.2. Characterization of Cu/CS Composite Hydrogel

In order to study the kinetics of the nanoparticles formation stabilized with chitosan and their reduction by the polymer in the obtained hydrogels, the UV-vis spectra of the composite materials were taken. The appearance of a surface plasmon resonance peak (SPR) at 685 nm indicates the formation of copper nanoparticles. It should be noted that the SPR peak of Cu usually lies in the range of 550–600 nm. The red shift in the SPR peak in the present case may be due to the formation of both Cu as well as Cu oxide nanoparticles [55,56], and its symmetry specifies that the synthesized nanoparticles have a spherical shape [57]. The dependence of position and intensity of the SPR peak on the exposure time of the composites was also studied (See Figure 2b). It was found that a noticeable reduction takes place over some time and includes the phase of the formation of some intermediate complexes in about a week after the preparation of the composites. Indeed, one can see in Figure 2b that during the first week the SPR peak position is shifted to a short-wave region, and the intensity of the peak also increased noticeably, which may indicate the increase in the copper nanoparticles size [58]. After the next week, the peak was fixed at 653 nm, while its intensity decreased, and then it began to slowly increase again. This increase probably indicates an increase in the amount of reduced nanoparticles [59]. The mentioned intermediate complex can be a form of a short-lived neutral cluster of copper nanoparticles or the so-called clusterite, which is a complex consisting of copper atoms and ions in a conjugated chain [60]. The latter structures can be destroyed if the optimal ratio of the number of atoms and metal ions is violated. The observed complexes can also be the clusters of several nanoparticles, which can subsequently separate into single ones [60]. The occurrence of intermediate complexes was observed by the authors of another article [61], who found that the formation and destruction of the intermediate complexes depends on controlling the amount of stabilizing and reducing agents.

Thus, over time, chitosan can partially reduce Cu^2+^ ions to metal nanoparticles. However, it should be noted that, despite the presence of copper nanoparticles in the composite, their amount is not large, as evidenced by the rather low intensity of the SPR peak, as well as the blue color of the hydrogel, indicating the presence of many Cu^2+^ ions in the composites.

A transmission electron microscope was used to visualize the morphology and the size distribution of copper nanoparticles in composite chitosan hydrogels (see Figure 3). The small nanoparticles (on average, 2–3 nm in diameter) of predominantly spherical shape were registered. It should be noted that the polydispersity of the nanoparticles obtained in this study is significantly lower than the observed spread in the sizes of copper nanoparticles stabilized with chitosan in traditional acidic solutions and subsequently reduced by NaBH_4_, prepared at normal atmospheric pressure. Moreover, the particles themselves are significantly smaller (for example, the diameter of the copper nanoparticles in article [62] ranges from 20 to 30 nm, in [63]—from 1 to 40 nm, in [64]—about 120 nm). Perhaps this is due to the significantly lower aggregation of chitosan in carbonic acid solutions under high CO_2_ pressure [35] compared to acetic acid solutions at normal pressure [65]. Reduced aggregation leaves more functional groups available to create a complex with metal ions, which leads to a better stabilization of the nanoparticles.

To confirm the UV-vis spectroscopy data, we compared TEM-microphotographs and histograms of particle size distribution in hydrogels with exposures of 15 min, 1, 3, and 6 weeks, respectively. Nothing was found in the first sample; most likely, the nanoparticles that cause the presence of a low-intensity SPR peak in UV-vis spectroscopy experiment were too small to be recorded by TEM. In the sample with 1-week exposure, in addition to copper nanoparticles, rather large complexes with a size range from 10 to 15 nm were found. Probably these are the complexes that are responsible for the high intensity of the 1-week SPR peak registered by the UV. No such agglomerates were found in other samples with longer exposure times. It should be noted that the size of nanoparticles decreased with exposure time. Considering the dynamics of the position and intensity of the SPR peak, it can be assumed that its initial increase is associated with the formation of nanoparticles and larger intermediate complexes, and the subsequent increase is associated with an increase in the number of nanoparticles related to stabilizing properties of chitosan.

X-ray photoelectron spectroscopy analysis was performed in order to gain more information about the specific interactions of Cu nanoparticles with chitosan dissolved in the solutions of carbonic acid under high CO_2_ pressure, as well as about the amount of the reduced nanoparticles.

There are several possible mechanisms describing the interaction of copper ions with chitosan. It is believed that, besides amino groups, hydroxyl groups are also involved in sorption, in the C-3 position most likely. Cu^2+^ can bind to several amino groups of the same chain or different chains via intramolecular or intermolecular complexation [66]. It has been found that the type of complexes capable of forming copper ions with functional groups depends on the pH of the solution. For example, coordination complexes such as [CuNH_2_(OH)_2_H_2_O] can form at pH 5.3–5.8, while [Cu(NH_2_)_2_(OH)_2_] can form at pH above 5.8 [67].

Figure 4a,b shows the C 1s spectra of the original chitosan and the composite with copper that reacted for 1 week with chitosan dissolved in carbonic acid under high CO_2_ pressure. The curve-fitting procedure was based on reliable chemical shifts [68].

According to Appendix A, there is a decrease in the relative intensities of the C-OH and NH_2_ groups. Such changes in the spectra may indicate the formation of a complex with copper ions, which involves the hydroxyl and amino groups of the polymer. Furthermore, a significant (approximately 2-fold) increase in the relative intensity of the peak corresponding to C-C/C-H groups is observed, which may indicate a certain redistribution of the charges of the polysaccharide backbone, which occurs as a result of the interaction of -OH and -NH_2_ groups with Cu. An increase in the relative intensity of the peak corresponding to the N(O)C group by 0.4 is also observed, which may indicate that the NH group of chitin units is also involved in the interaction of chitosan with copper. A similar effect was observed for a composite based on chitosan dissolved in the presence of carbonic acid and silver [69]. An increase in the relative intensity of the peak corresponding to the NH_3_^+^ group indicates that a certain number of chitosan amino groups protonate after exposure in carbonic acid solutions.

Using the X-ray photoelectron spectra of Cu 2p_3/2_ (Figure 4d), we calculated the percentage of copper oxide and reduced metal atoms in the hydrogel composite formed on the basis of chitosan in a solution of carbonic acid under high pressure CO_2_ and a solution of copper sulfate after a week of exposure. It turned out that the percentage of reduced metal in the sample is only 3%. This is also indirectly indicated by the color of the gel, which remained blue even after several months of exposure. At the same time, a comparison of E_b_, W, I_rel_ and peak-shape information for these samples with that of standard sample values [70,71] shows that the existing in the sample Cu^2+^ state may be related to a mixture of CuO and Cu(OH)_2_. According to [71] I(II)/I(III) intensity ratio (where I(II) and I(III) are relative intensities of peak II and III, respectively) in the Cu 2p_3/2_ spectrum for CuO is 0.94 (31/33), while peak III is not observed in the spectrum for Cu(OH)_2_. Therefore, I(II) − I(III) × 0.94 may be a measure of Cu(OH)_2_ content, which turned out to be 7% in the composite with copper, which reacted for 1 week with chitosan dissolved in the presence of carbonic acid under high CO_2_ pressure.

Further, in this work, to reveal the nature of interactions between chitosan units and copper nanoparticles, IR spectroscopy of pure chitosan and composite hydrogels with 1- and 6-week exposures was performed.

As can be seen from the graphs presented (see Figure 5), the peak of the initial chitosan at 3444 cm^−1^ responsible for the overlapping stretching of –OH and –NH_2_ involved in the formation of intermolecular hydrogen bonds [72,73], noticeably broadens in the spectra of all Cu/CS composites (hereinafter, see Appendix A). This indicates the participation of hydroxyl groups in the process of coordination complex formation. In the spectra of hydrogels, the relative height of this peak (normalized to the peak responsible for –CH symmetric bending vibrations in CHOH, which is at 1379 cm^−1^ [74] and not involved in the formation of the complex) slowly increases with hydrogel exposure time. That can be explained by the new nanoparticles number increase and chitosan complexation with Cu^2+^ ions, and, consequently, the participation of an increasing number of –OH and –NH_2_ groups in composite formation. The peak in the spectrum of pure chitosan at 1588 cm^−1^, corresponding to deformation vibrations of amino groups [72], in the spectra of Cu/CS composites is shifted to the long-wavelength region of the spectrum, and its relative height increases with hydrogel exposure time. This can be regarded as an additional confirmation of the interaction of Cu^2+^ ions with chitosan amino groups. Furthermore, it can be observed that relative intensities of C-OH stretching vibration peaks at 1095 cm^−1^ decrease for both Cu/CS composites compared to the spectrum of pure chitosan, this tendency was also observed previously [31]. This may also indicate the involvement of hydroxyl groups in the complex formation of chitosan with copper nanoparticles. Thus, the IR spectroscopy data reveal the participation of amino and hydroxyl groups in formation of the Cu/CS coordination complexes.

### 2.3. Antimicrobial Activity

It is known that copper nanoparticles have a pronounced antimicrobial activity [2], therefore experimental studies of the activity of the obtained chitosan-copper composites against fungi and bacteria were carried out (see Table 1). Dried Cu/CS thin films exhibited a pronounced antibacterial activity against both Gram-positive bacteria (*B. subtilis*) and yeast fungi (*C. albicans*).

In addition to the antimicrobial effect of copper, it should be taken into account that chitosan, which has a positive charge, can bind and interact with the negatively charged surface of a bacterial cell, increasing the membrane permeability and causing the death of bacteria. In addition, chitosan can bind to DNA and inhibit mRNA synthesis [75].

### 2.4. Porous Gel Modification

As is known, the porosity of chitosan-based lyophilized gels crosslinked with heavy metals is very limited and such gels typically have a smooth surface structure [76]. Many techniques have been suggested in the literature to control the pore sizes distribution in chitosan membranes using chemical and physical modifications [77]. To modify chitosan-based membranes, silicon dioxide or polyethylene glycol is often used as one of the components of the composite [77,78]. Reaction with ethylene oxide, potassium hydrogen phthalate and glutaraldehyde, including other derivatizations and copolymerization are the most common chemical techniques. However, all these chemical routes also introduce changes in hydrophilicity and permeability of the membrane as well as introduce new components into the composite, which is not always desirable [79,80,81].

We found that if hydrogels are saturated with a mixture of H_2_ + CO_2_, or He + CO_2_ at a total pressure of 30 MPa, then during decompression a significant foaming of the hydrogel occurs that is easy to see (Appendix A). In Figure 6 one can see more detailed SEM micrographs of the inner structure of the lyophilized modified gels saturated with a mixture of H_2_ + CO_2_, or He + CO_2_ and only with CO_2_ or H_2_ for comparison. The results of the porosity and density measurements of the samples are given in Table 2. It turned out that such a modification of the gels significantly reduces the characteristic pore size and increases the porosity. Firstly, the effect of increasing the solubility of gases such as H_2_ in class 3 media in gradation according to the ability to dissolve CO_2_ (these include ionic liquids, polymers, oil) is known [82,83]. Thus, it can be assumed that dissolved CO_2_ under high pressure in our case also increased the solubility of hydrogen and helium in the bulk of the swollen chitosan hydrogel. Secondly, during decompression, due to the very high diffusion coefficient of H_2_ and He [84], these gases tend to quickly leave the polymer volume much faster than CO_2_, creating supersaturation faster and forming many nuclei of pores of a much smaller size than in the case of pure CO_2_. For scaffolds based on chitosan with a degree of deacetylation of 75–85%, there are pores in the range of 30–150 μm [85]. For the porous collagen/chitosan composite scaffold, the mean pore size increased from 100 μm for the uncrosslinked composite, to more than 200 μm for the scaffolds crosslinked with glutaraldehyde [86]. In general, the pore size in different chitosan-based scaffolds varies from 20–100 μm for a chitosan-alginate-nSiO_2_ scaffold [87] to 400–600 μm for a pure chitosan scaffold [87]. As one can see, using our modification method, porous composites with low degree of polydispersity can be obtained. The porosity of different scaffolds varies from around 30% for pure chitosan to 90% for chitosan/alginate composite [87]. For the chitosan–protein scaffolds [88], the porosity turned out to be about 26%. The densities of pure chitosan film typically range from about 1.1 g/cm^3^ [89], to 1.22 g/cm^3^ [90], and up to about 1.4 g/cm^3^ for both low and high molecular weight chitosan films [91], while for a chitosan-based scaffold it can vary from 0.3 g/cm^3^ [92] to 1.2 g/cm^3^, which is for TiO_2_/chitosan scaffold [93]. Thus, we see a noticeable decrease in the pore size and density as a result of modification of Cu/CS composites by the method described above. Due to the mentioned antimicrobial effect of Cu/CS composites, such microporous sponges can be used as filters, in regions where the water is heavily contaminated with bacteria.

It was found in the work [94] that CO_2_ and H_2_ are able to penetrate through the chitosan membrane at room temperature, and also it was shown that the permeability of CO_2_, H_2_, and N_2_ gases at 0.15 MPa in the dried chitosan membrane decrease with increasing temperature (the experiment was carried out in the temperature range from 20 to 150 °C), as does the diffusion ability. Thus, the temperature regime in our studies can also contribute to the better penetration of CO_2_, H_2_ and He into the chitosan matrix. For some polymers, it has been shown that the solubility of CO_2_ in the polymer increases with increasing pressure [95]. It should be noted that the diffusion coefficient of H_2_ and He, other things being equal, is higher than that of CO_2_ for a wide range of different polymers [84], which further explains the combined effect of the gases in our experiment. The gas permeability in another cationic polymer poly(4-vinylpyridine) was tested for a number of different gases and it decreased in the order of gases: H_2_ = He > CO_2_ > O_2_ > CH_4_ > N_2_, while the gas diffusivity and solubility in poly(4-vinylpyridine) film decreased in the order of gases: He > H_2_ > O_2_ > CO_2_ = N_2_ > CH_4_ and CO_2_ > CH_4_ > O_2_ > N_2_ > H_2_ > He, respectively [96]. Thus, indeed, the formation of small pores (nuclei) in gels is due to the high diffusivity of H_2_ and He, while the penetration of these gases into the composite is promoted by CO_2_.

### 2.5. Characteristics of Hydrogel Spheres

It turned out that if small amounts of copper sulfate aqueous solution are added dropwise to a solution of high molecular weight chitosan in aqueous carbonic acid, stable elastic hydrogel spheres can be obtained. Using low molecular weight chitosan, such well-defined geometric shapes cannot be obtained because of the lower viscosity of chitosan solution.

The possibility to obtain spherical hydrogels of a given size allows them to be effectively used for catalysis, since such composites will be more active than gel films previously described in the literature [31], due to their accessible shape and increased surface area. The mass distributions of the spherical hydrogels obtained are shown in Figure 7. The degree of their polydispersity decreases with an increase in the size of the spheres: the deviation from the average mass does not exceed five percent for spheres prepared by dropping 100 μL of chitosan solution; for samples obtained with 25 and 50 μL, the deviation is approximately ten percent. However, it should be taken into account that the larger the volume of the injected chitosan solution, the more the shape of the composite differs from the spherical one. Therefore, the method described in this paper for the preparation of hydrogel spheres with a given volume of the substance contained in them is quite effective for relatively small composites.

It should be noted that the idea of creating such hydrogel spheres is known. Previously, similar composites have already been created by some authors in traditional solvents. For example, in the article [97], for the preparation of porous composites with aurum nanoparticles, chitosan was first dissolved in an acidic medium and then gelled in an alkaline solution. The resulting hydrogels were immersed in a HAuCl_4_ solution to form a composite, after which the Au/CS hydrogel microspheres were immersed in aqueous ethanol solutions, then in a NaBH_4_ solution, and finally freeze-dried. In another work [98], a multistage method for the preparation of microspheres was used, which consisted of the solvothermal formation of copper NPs from Cu(NO_3_)_2_ dissolved in ethylene glycol and reduction in situ at 150 °C.

The advantages of the composite preparation method described in this article are its simplicity and ease of implementation, as well as the absence of any impurities in the spheres except for chitosan and Cu nanoparticles formed from copper sulfate.

The internal structure of the hydrogel spheres was studied using SEM and is clearly visible in the micrographs (see Figure 8a,b): at the interface between two media, where the hydrogel shell is formed, there is a large number of Cu-containing crystalline complexes to be observed. Their number decreases towards the center of the sphere, where the layers are observed, which are standard for a lyophilized solution of chitosan. Between the outer shell and the inner part of the sphere containing mainly chitosan, there is an area with pronounced pores, elongated ellipsoidal and spherical in shape, approximately from 8 to 40 μm in length and from 8 to 20 μm in width. The sizes of crystals present in the composite do not exceed 1 µm (see Figure 8d). The evidence that the crystals shown in the SEM micrographs are Cu-containing is the presence of well-defined peaks in the IR spectrum (see Figure 8c) of the composite at wavelengths of 418 and 668 cm^−1^, which corresponds to CuO (Cu(I)–O vibration) and Cu_2_O (Cu(II)–O vibration), respectively [99]. It should be noted that similar crystals were also observed by the other authors. For example, in the work [100] a small amount of Cu_2_O crystals formed as a result of the oxidation of metallic copper was recorded. A possible type of reaction is given below:2Cu^0^ + H_2_O_2_ → Cu_2_O + H_2_O.

Hydrogen peroxide could be formed as a result of the influence on water with some external stimuli, because the spontaneously formed H• and OH• radicals readily recombine even in the absence of any additives or scavengers producing H_2_, H_2_O_2_, and H_2_O. The authors note that when the experiment was carried out in an atmosphere of argon and hydrogen, only metallic copper nanoparticles were obtained. Probably, in our case, Cu^+^-containing crystals are formed after partial reduction of copper from the divalent to a monovalent state.

According to the EDX analysis, the ratio of Cu/CS in the sample can be calculated (See Appendix A). The great advantage of the obtained spherical hydrogels is that the copper nanoparticles are concentrated mainly on the surface of the sphere and are easily accessible for possible reagents.

### 2.6. Properties of Hydrogel Spheres

Composites were tested for swelling capacity under various conditions, namely: swelling ability in an aqueous medium depending on time, in acidic and alkaline media at various pH values and with temperature increase. The results of the experiments are shown in Figure 9.

The swelling curves of the freshly prepared gels in water indicate a decrease in the gels mass, and then, in 40 min, the value reaches a plateau (see Figure 9a). This can be explained by the fact that the formation of spheres takes some time, and the macromolecules of the network framework of the sphere must reach the most favorable possible conformation, which is accompanied by a decrease in the internal volume of the capsule and, consequently, free space for water molecules, after which the spheres remain in a stationary state.

As the temperature increases, (see Figure 9c), the samples significantly lose mass. This is probably due to the destruction of bonds between copper nanoparticles and chitosan macromolecules, which leads to the degradation of the gel shell. This is also due to the fact that the composites we studied do not contain any additional crosslinking agents, as a result of which polymer chains remain flexible and an increase in temperature can cause the breaking of secondary interactions [101].

The results of the experiment on the behavior of the spheres at different pH values are shown in Figure 9d. It should be noted that a noticeable decrease in the size of the spheres was recorded (from about 3 mm in a neutral medium to about 2 mm at pH 13). Such changes can be explained by the fact that in an alkaline medium the amino groups of chitosan are not charged, in contrast to the state in acidic medium, where the amino groups are protonated and an additional electrostatic repulsion occurs, due to which the gel expands and absorbs more water. At high pH, the crosslinked polymer chains are in the so-called poor solvent, which causes the polymer network to collapse. In addition, the color of the spheres became dark blue, which can be explained by the formation of a stable [Cu(OH)_4_]^2−^ anion. Since the spheres do not degrade, it can be assumed that the exchange reaction with the formation of the mentioned anions occur only on the surface of the sphere (probably, because of concentrated CuSO_4_ solution, some Cu^2+^ ions, coordinated with chitosan, have an ability to bond with extra groups, for example, OH- from alkaline medium), and the bulk internal part of the hydrogel remains intact, due to which the structure is preserved. This assumption is confirmed by the color of the spheres in a solution with pH 12, when the violet color has not yet reached its maximum intensity: only the outer part of the composites changed color (see Appendix A). A rather sharp change in mass, size and color occurred only at pH 12, which indicates the high stability of the original hydrogel system.

To gain the insight what happens to composite spheres at pH 13, an XPS study was conducted. As one can see from the Appendix A, in the initial as-prepared sample there is significant amount of S, which may appear due to the presence of residual sulfate groups on the surface of the sphere, attached to the Cu nanoparticles. At the same time, XPS data for a sphere exposed to NaOH solution at pH 13 showed the complete disappearance of sulfur.

This may indicate the replacement of the sulfate group by the hydroxyl group. In this case, the presence of residual Na is detected in such a sample, which could appear as a result of their coordination interactions with composite electronegative atoms.

A comparison of the Cu 2p spectra for the as-prepared composite chitosan spheres with Cu and the same spheres at pH 13 shows the absence of CuSO_4_, since the characteristic satellites at ~940 and 960 eV are absent (see Appendix A). At the same time the content of Cu(OH)_2_ in the Cu^2+^ state increased to the value of 23%.

### 2.7. Catalytic Activity

To study the catalytic activity of the freeze-dried composite chitosan capsules with copper nanoparticles, the reduction of nitrobenzene to aniline in the presence of NaBH_4_ was carried out. It should be noted here that such a 3D form of a chitosan composite with copper nanoparticles was obtained for the first time. Moreover, due to the fact that such capsules were prepared on the basis of chitosan dissolved in self-neutralizing carbonic acid, they can be considered as especially pure reagent, which does not contain any impurities that could potentially poison catalysis. In each reaction, six lyophilized composite chitosan capsules with copper nanocrystals were used in order to create more active centers with a view to speed up the reaction (the diameter of one capsule is approximately 2 mm).

Using elemental analysis, the amount of copper in each capsule was calculated. The results are presented in the Appendix A. This way, in each reaction approximately 0.8 mg of catalyst participated (molar ration nitrobenzene/NaBH_4_/Cu = 1/2/1).

A number of chemical and biological methods have been developed that are used to reduce nitrobenzene to aniline, such as microelectrolysis [102], electrochemical reduction [103], and the biological anaerobic process [104]. Typically, chemical methods can reduce nitrobenzene faster than biological processes, but they require more or significant amounts of chemicals and relatively expensive noble and transition metal catalysts, which add cost and can cause secondary pollution. However, biological methods, while being more efficient from an economic point of view, provide a slower reaction rate [105]. In addition to the need for industrial production of aniline, the problem of reducing the content of nitrobenzene is also associated with its toxicity. This compound belongs to the second hazard class and in high concentrations can cause hemolysis [106]. Absorbed through the skin, nitrobenzene has a strong effect on the central nervous system, disrupts metabolism, causes liver disease, oxidizes hemoglobin to methemoglobin. In experiments in experimental rats and mice, after inhalation of nitrobenzene vapors, toxic damage to the liver, lungs and spleen was observed. Some governments [107,108] have imposed strict limits on environmental concentrations of nitrobenzene, such as 17 µg/L in the US and 20 µg/L in China. As noted earlier, nitrobenzene is a toxic compound that is often encountered as an environmental pollutant, while aniline is a less toxic end product (aromatic amines are 500 times less toxic) that mineralizes much more easily [100].

The UV-vis spectra of the reaction mixture showed noticeable changes along over time (see Figure 10a). The intensity of the absorption band corresponding to nitrobenzene (near 265 nm) gradually decreased, while the absorption band corresponding to aniline (near 230 nm) appeared [31]. After the reaction time (which took only about 25 min), the nitrobenzene peak almost disappeared, which confirms the successful conversion reaction. It was found that at the end of the process, the spheres can be retrieved from the reaction media, washed in distilled water, freeze-dried and used again (see Figure 10b).

It should be noted that the swelling curves of the freeze-dried hydrogels look different than fresh ones, as it was also observed in work [31]. According to the experimental graph (see Figure 9b), at first the composite mass increases, and then, after 25 min, remains stable. This behavior of the composite capsules can be considered as an advantage for the catalysis purposes, since it should provide fast solvent inflow to the Cu nanoparticles located inside the pores of the lyophilized sphere, and easy access to the nanoparticles for the substances participating in the reaction and ensure its stability during the entire reaction time required for its completion.

Calculated by the method described in Section 3.7, the yield of the product in the nitrobenzene reduction reaction does not drop below 67% for at least five cycles (see Figure 10b). We compared these results with those described in the literature for catalysts based on copper nanoparticles. It should be noted that the reaction time itself was about 25 min at room temperature, which is much faster than in some other works. For example, the authors of work [109] found that it is possible to accelerate the reduction reaction to 2 h at a temperature of 30 °C (molar ratio Nitrobenzene/NaBH_4_/Cu = 1/3/0.1) with a yield about 82%. In the work [110], copper nanoparticles were used to reduce nitrobenzene to aniline at a temperature of 80 °C, the reaction proceeded in several minutes with a 100% yield (molar ratio nitrobenzene/NaBH_4_/Cu = 1/2/0.15). In the work [31], after 60 min of reaction at room temperature, a yield of about 90% was obtained using a chitosan film with copper nanoparticles as a catalyst (molar ratio Nitrobenzene/NaBH_4_/Cu = 1/2/40). Thus, the catalyst, proposed in this work, is potentially competitive with respect to those that already described in the literature.

## 3. Materials and Methods

### 3.1. Materials

In the present research, chitosan powders supplied by Sigma-Aldrich (Saint Louis, MO, USA) (molecular weights M_1_ = 210 kg/mol, M_2_ = 1300 kg/mol; degrees of deacetylation DD_1_ = 80%, DD_2_ = 70% [35]), high purity carbon dioxide (99.995%, Moscow Gas Refinely Plant, Moscow, Russia), specially purified and deionized water (Millipore Milli-Q, MilliporeSigma, Burlington, NJ, USA), copper sulfate (cat. No. 451657, Sigma-Aldrich, Saint Louis, MO, USA), sodium borohydride (cat. No. 16940-66-2, Acros organics, Geel, Belgium), nitrobenzene (cat. No 98-95-3, Sigma-Aldrich, Saint Louis, MO, USA), hydrogen (99.99%, Gazprodukt, Moscow, Russia), and helium (99.995%, Gazprodukt, Moscow, Russia) were used.

### 3.2. Gel Preparation

A detailed description of chitosan dissolution process in carbonic acid can be found in our previous work [38]. Aqueous solution of CuSO_4_ 0.012 M or 0.024 M (2.5 mL) was added to the chitosan solution (5 mL) to achieve the ratio Cu/chitosan = 1/10 or 1/5, respectively (the pH of the mixture after the addition of Cu^2+^ was measured by pH-indicator strips and was found to be about 6.5). After that, the solution was mixed with a spatula until a hydrogel was formed (see Figure 1a). The resulting Cu/CS composite was stirred with a spatula to a state characterized by a noticeable increase in the viscosity of the system. The gel formation was also confirmed by rheology tests. Then it was kept for aging in a glass flask in a dark place at room temperature.

### 3.3. Micrporous Gel Preparation

To modify the hydrogel structure, the Cu/CS composite based on low molecular weight chitosan in wet state was immersed in the reactor, then the reactor was hermetically sealed, and hydrogen and carbon dioxide were successively pumped into it through a valve until pressures of 1 and 30 MPa, respectively, were reached. After a week of exposure (this time was chosen to make sure that all diffusion processes were completed and the system reached a steady state), the reactor was depressurized with a decompression rate of 1 MPa/min; a microporous hydrogel was obtained and lyophilized for further experiments as described below.

The porosity of lyophilized gels was determined by filling the free volume inside the sample with a liquid using 95% ethanol. The volume and weight of a dry sample were measured, then it was immersed in a vessel with ethanol for five minutes. After that, the sample was taken out, excess ethanol was removed with filter paper, and then the sample was weighed. Porosity was calculated by the formula:(1)$$\varphi =\frac{m-m0}{\rho \times V}\times 100\%,\;$$
where m is the mass of the sample after soaking in ethanol, m0 is the mass of the dry sample, V is the volume of the sample, which was calculated from its linear dimensions, ρ is the density of ethanol.

The density of lyophilized gels was calculated gravimetrically. First, the dried gel was weighed. Then the sample was fixed on the wire and immersed in ethanol, and then the readings of the balance, which was previously reset to zero, were recorded.

The density of the sample was calculated by the formula:(2)$$\rho 0=\frac{m0}{m0-\Delta m}\times \rho ,$$
where m0 is the initial mass of the sample, Δm is the balance readings, ρ is the density of ethanol.

### 3.4. Synthesis of Cu/CS Spherical Composites

A solution of high molecular weight chitosan was added in drops (25, 50, 100, 200, or 300 µL) to aqueous solutions of CuSO_4_ (0.116 M) using a mechanical pipette. Next, the solution with a drop was shaken for several minutes, after which the resulting hydrogel sphere was removed (See Figure 1b). Subsequently, the prepared spheres were stored in water.

To study the swelling kinetics of the spheres in water, freshly prepared composites and freeze-dried ones prepared with chitosan solution volume of 100 μL were taken.

To study the behavior of the freshly obtained spheres in an alkaline media, NaOH solutions were prepared with pH from 8 to 13 in increments of 1. The spheres were kept in each of the solutions for 1.5 h. After each experiment, the composites were removed from the solutions, the excess liquid on the surface was blotted with filter paper, and then composites were weighed.

The amount of Cu in in the samples with different amounts of introduced chitosan was obtained using the X-ray fluorescence method on a spectroscan MAKC-GVM (Spektron, Saint Petersburg, Russia) with X-ray tube, a Pd anode (40 kV, 0.5 mA), a proportional sealed-off detector, and a LiF200 crystal analyzer. The measurements were carried out in the air.

Briefly, samples preparation consisted of taking micro-weights of about 5 mg of crushed freeze-dried composites (Mettler Toledo XP6 microbalance, Mettler-Toledo AG, Greifensee, Switzerland) and 980–990 mg of polystyrene (PSE-1) (Mettler Toledo AB 265-S balance, Mettler-Toledo AG, Greifensee, Switzerland). Weighed samples and polystyrene were transferred into agate mortars, then, 20 mL of ethanol was added, and the system was thoroughly mixed. After that, it was dried in the air and formed into tablets with a diameter of 2 cm under a pressure of 10 tons. The dilution factor was calculated by the formula:(3)$$K=\frac{m\left(PS\right)+m\left(\mathrm{Cu}/\mathrm{CS}\right)}{m\left(\mathrm{Cu}/\mathrm{CS}\right)},$$
where m(PS) is the mass of polystyrene, m(Cu/CS) is the mass of the studied composite.

It should be noted that the test sample can be unevenly distributed over the tablet, as a result of which the intensity measurements were made twice for each side of the tablet. Determination of the copper content in the tablet was made using the analytical program on spectrometer.

The copper percentage content in the Cu/CS spherical composite was calculated by the formula:(4)C=C0×K 
where C0 is the copper content in the tablet.

### 3.5. Characterization

Rheology analyses were performed on an Anton Paar MCR 302 rotational rheometer (Anton Paar GmbH, Graz, Austria) at 20 °C. A measuring cell with plate–plate configuration was used. Disc-shaped samples, 25 mm in diameter and about 4 mm thick were placed between stationary bottom plate and upper plate connected with a rotor. Diameter of the upper plate is 25 mm. The gap between lower and upper plate was set so that full contact between upper plate and top surface of the sample was visible. Measurements were carried out at the room temperature in the dynamic mode of forced torsion oscillations at a constant strain amplitude of γ = 1%. The oscillation angular frequency *w* was varied in the range 0.628–250 rad/s.

FTIR spectra of thin composite films obtained by drying of Cu/CS hydrogels in air at 35 °C were registered on a Nexus FTIR 470 (Thermo Nicolet, Waltham, MA, USA) spectrometer equipped with a Transmission accessory in the range of 4000–400 cm^−1^ and an attenuated total reflectance (ATR) accessory with ZnSe crystal in the range of 4000–650 cm^−1^. FTIR spectra were processed using OMNIC (Thermo Nicolet) software. X-ray photoelectron spectra of air-dried composite films were acquired with an Axis Ultra DLD (Kratos, Stretford, UK) spectrometer using monochromatized Al Kα (1486.6 eV) radiation at an X-ray tube operating power of 150 W. Survey and high-resolution spectra of appropriate core levels were recorded at pass energies of 160 and 40 eV and with step sizes of 1 and 0.1 eV, respectively. The surface charge was taken into account according to the C–OH state identified in the C 1 s spectrum, to which a binding energy of 286.73 eV was assigned [68]. In the case of CuSO_4_ the sample charging was corrected by referencing to the C–C/C–H state (284.8 eV). UV–vis spectra of Cu/CS hydrogels synthesized directly in quartz cuvettes were obtained by means of a Unicam Helios Alpha spectrometer (Unicam, Dowlish Ford, UK).

For the SEM study, the gel was first frozen in a freezer at −18 °C, then in liquid nitrogen, and lyophilized using a freeze-dryer Alpha 1–2 LD (Martin Christ, Osterode am Harz, Germany) at −50 °C and 0.04 mbar for 2 days. The lyophilized gel was cleaved in the liquid nitrogen immediately before the SEM study. SEM study was carried out using a Hitachi SU8000 field-emission scanning electron microscope (FE-SEM) and a Hitachi TM4000 (Hitachi Ltd., Tokyo, Japan) scanning electron microscope. Before measurements, the samples were fixed on an aluminum holder with a conductive carbon tape, and some of them were coated with a 15 nm film of carbon. The images were acquired in a backscattered electron mode at 10 kV accelerating voltage. EDS-SEM studies were carried out using an X-max 80 EDS system (Oxford Instruments, Abingon, UK) at 20 kV accelerating voltage. Target-oriented approach was utilized for the optimization of the analytic measurements. For the TEM study, a sample of Cu/CS hydrogel was dried into a film in a heating cabinet (were oven-dried) at 35 °C. Then, thin sections were made from it using an Ultracut-R ultramicrotome (Leica Microsystems GmbH, Mannheim, Germany) with an ultra 35° diamond knife (DiaTOME, Switzerland). They were transferred from the surface of the knife using a dry method to a carbon film-substrate model S160 (Plano GmbH, Wetzlar, Germany) fixed on a copper grid. TEM micrographs were taken on a LEO 912 AB OMEGA microscope (LEO/Carl Zeiss, Jena, Germany) with an accelerating voltage of 100 kV.

### 3.6. Antimicrobial Activity

The antimicrobial activity of chitosan composites with metal nanoparticles was determined by the disk diffusion method. Antifungal activity was assessed using test strain of yeast *Candida albicans* ATCC 2091. The antibacterial activity was studied using test cultures of Gram-positive bacteria strains—*Bacillus subtilis* ATCC 6633 from the collection of cultures of the Gause Institute of New Antibiotics (Moscow, Russia). Bacteria were cultured for 18 h in Luria broth (HiMedia, Thane West, Maharashtra, India), after which 50 µL of bacterial suspension (containing ~10^7^ CFU) was mixed with 10 mL of Luria agar (LBagar) and immediately placed in a Petri dish, which was pre-coated with 1.5% agar. Fungi were cultivated for 24 h on Potato-Dextrose agar (PDA agar) plates (Sigma-Aldrich, St. Louis, MO, USA). Test samples were transferred to the top layer of soft agar. After incubation at 37 °C overnight, sites of inhibition were observed. Disks of chitosan films 6 mm thick with metal nanoparticles were immersed in ethanol for 1 min, dried, and placed in a Petri dish. Inhibition zones were measured manually using a digital caliper. Assays were performed in triplicate. The sensitivity of the test organisms was controlled with standard discs containing amphotericin B (40 µg/disc) for yeast fungi and amoxiclav/clavulonic acid (20/10 µg/disc) for bacteria.

### 3.7. Model Catalysis Reaction

To test the catalytic activity of the Cu/CS spheres, the model reaction of the nitrobenzene reduction to aniline was carried out. An aqueous solution of 5 × 10^−3^ M nitrobenzene and a freshly prepared aqueous solution of NaBH_4_ 50 × 10^−3^ M were used. An amount of 2 mL of an aqueous solution of nitrobenzene, 0.5 mL of an aqueous solution of NaBH_4_, and six dried gel spheres (0.016 m, 100 μL) were added to an optical cuvette for UV-vis spectroscopy measurements (Unicam Helios Alpha, Unicam, Dowlish Ford, UK). The system was kept static at room temperature. After each reaction, composites were rinsed with water and dried for further using. An analytical curve was built to calculate the conversion efficiency. The absorbances of nitrobenzene solutions with concentrations 0.1, 0.2, 0.4, 0.6, 0.8, 1, and 2 × 10^−3^ M were measured by a UV-vis spectrometer. After each reduction reaction (25 min), the concentration of nitrobenzene remaining in the reaction system was calculated using the analytical curve. The conversion efficiency was calculated using the formula:(5)$$\eta =\frac{C0-Ce}{C0}\times 100\%,$$
where C0 is the initial concentration of nitrobenzene and Ce is the nitrobenzene concentration after the reduction reaction.

## 4. Conclusions

In this study, it was shown for the first time that chitosan dissolved in water saturated with CO_2_ under high pressure is able to stabilize and partially reduce Cu^2+^ ions to small-sized, predominantly spherical metal nanoparticles with a low degree of polydispersity, about 2 nm in diameter. At the same time, it was found that a sufficiently stable elastic polymer hydrogel could be formed. Moreover, its shear modulus was approximately 8 times greater than loss modulus and this ratio was maintained over the time. It was found out that the reduction of Cu nanoparticles with chitosan includes the stage of formation of some intermediate compound, which does not affect the spherical shape of the composites, but makes them slightly larger and somewhat worsens the rheological characteristics of the gel, but is short-lived and does not affect the final product. It has also been proven that the resulting composites have a fairly pronounced antimicrobial activity against fungi and bacteria. Moreover, in case of fungi, the antibacterial activity of Cu/CS composite film was greater than that of antibiotic. It is also worth noting that in the case of using these composites in the form of sponges as filters, for example, for water, their important property is the absence of the need for additional purification of such filters from the residual solvent, traces of which can additionally contaminate the medium being cleaned.

It was found that when the hydrogel is saturated with a mixture of H_2_ + CO_2_ / He + CO_2_ fluids under high pressure (1 MPa and 30 MPa, respectively), it is possible to reduce tenfold the pore size of the resulting material to several micrometers, while having low degree of polydispersity, compared to the known examples of chitosan porous matrices, and increase their porosity.

By varying the molecular weight of chitosan, we were able to change the 3D morphology of the gels, creating stable hydrogels, films, capsules, and porous lyophilized gels. Chitosan-based capsules crosslinked with Cu nanoparticles were obtained for the first time and showed good temperature stability up to 55 °C and alkaline medium stability up to pH 11, as well as stability in aqueous solutions during the time that is important for their storage. At the same time, it was found that these capsules, when used as a reusable catalyst, make it possible to accelerate the reaction by two times up to 25 min using 40-times less metal in the reaction under standard conditions. The spherical shape of the composites made it possible to increase the specific surface for catalytic activity magnification. Another advantage is that such catalysts can be easily removed from the reaction media without any additional purification of the solution. We would also like to note once again that this composite, obtained using a new “green” chitosan solvent, is pure catalyst, which does not need to be further cleaned.

The proposed system with flexible properties can be useful in biomedical applications, as filters or catalysts.

## Figures and Tables

**Figure 1 molecules-27-07261-f001:**
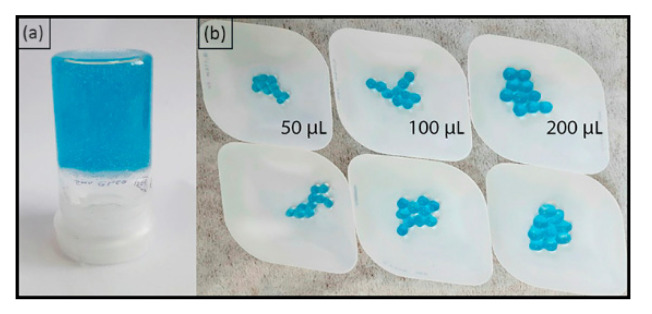
(**a**) Cu/CS hydrogel sample made from low molecular weight chitosan; (**b**) Cu/CS spherical hydrogel samples fabricated from high molecular weight chitosan with different volumes of added chitosan solution.

**Figure 2 molecules-27-07261-f002:**
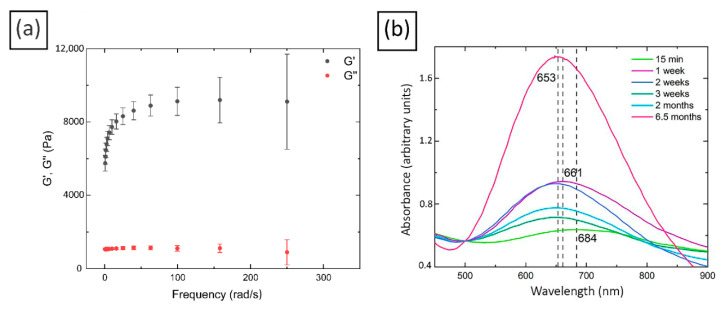
(**a**) Rheological study of freshly prepared Cu/CS composites (low molecular weight chitosan, molar ratio Cu/CS = 1/5); (**b**) UV-visible spectra of Cu/CS composites (low molecular weight chitosan, molar ratio Cu/CS = 1/10).

**Figure 3 molecules-27-07261-f003:**
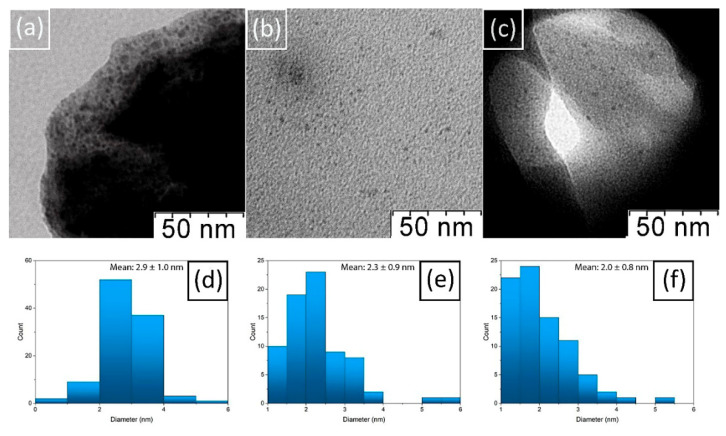
TEM-micrographs of (**a**) 1-week Cu/CS gel sample (low molecular weight chitosan, molar ratio Cu/CS = 1/5); (**b**) 3-weeks Cu/CS gel sample; (**c**) 6-weeks Cu/CS gel; Histogram of size distribution of nanoparticles (**d**) in 1-week Cu/CS gel sample; (**e**) 3-week Cu/CS gel sample; (**f**) in 6-weeks Cu/CS gel sample.

**Figure 4 molecules-27-07261-f004:**
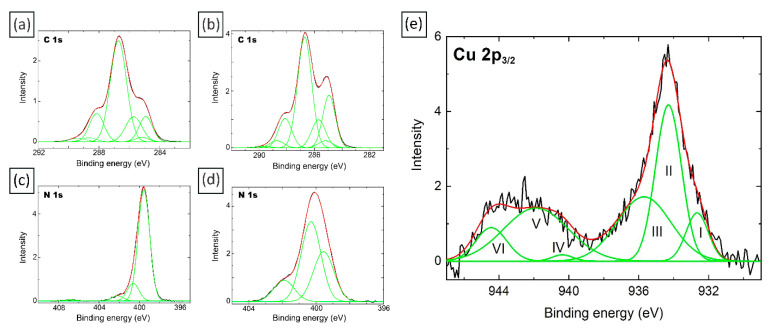
C 1s and N 1s X-ray photoelectron spectra of the original chitosan—(**a**,**c**)—and the 1-week Cu/CS composite—(**b**,**d**); (**e**) Cu 2p_3/2_ spectra (low molecular weight chitosan, molar ratio Cu/CS = 1/5). The roman numerals identify the numbers of peaks.

**Figure 5 molecules-27-07261-f005:**
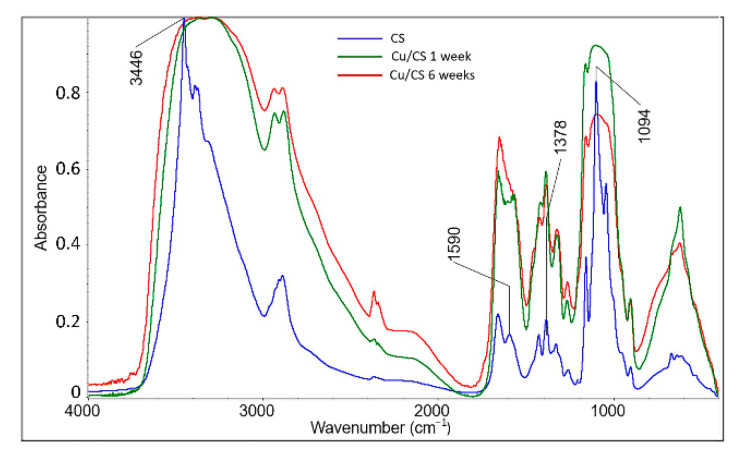
FTIR spectra of Cu/CS air-dried thin films (low molecular weight chitosan, molar ratio Cu/CS = 1/10) with different exposure times.

**Figure 6 molecules-27-07261-f006:**
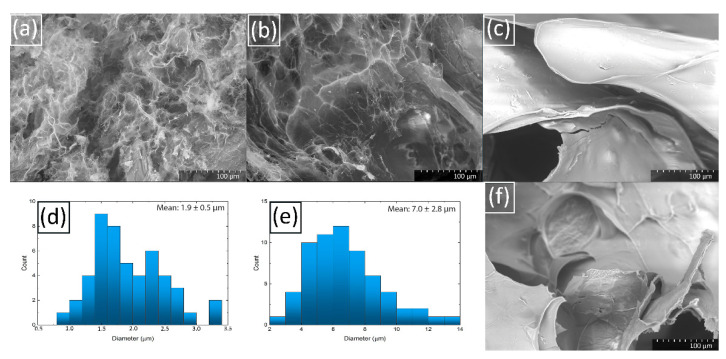
SEM-microphotographs Cu/CS gel samples (low molecular weight chitosan, molar ratio Cu/CS = 1/5) modified with (**a**) H_2_ (1 MPa) and CO_2_ (30 MPa); (**b**) He (1 MPa) and CO_2_ (30 MPa); (**c**) saturated with H_2_ (1 MPa); (**d**) Histogram of distribution of pore sizes in sample (**a**); (**e**) Histogram of distribution of pore sizes in sample (**b**); (**f**) SEM-micrograph Cu/CS gel sample saturated with CO_2_ (30 MPa).

**Figure 7 molecules-27-07261-f007:**
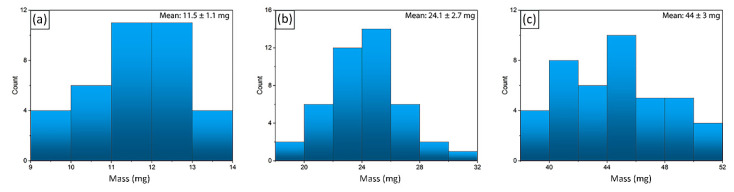
Histogram of mass distributions of Cu/CS spherical composites with (**a**) 25 μL; (**b**) 50 μL; (**c**) 100 μL of chitosan solution added in aqueous carbonic acid.

**Figure 8 molecules-27-07261-f008:**
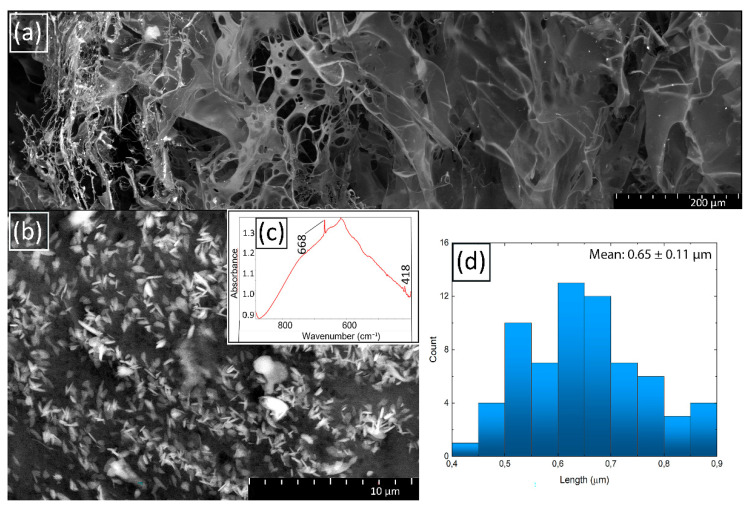
SEM-microphotographs of Cu/CS spherical gel sample (high molecular weight chitosan, 0.116 M, 300 µL) (**a**) cleavage; (**b**) surface; (**c**) part of FTIR spectra of Cu/CS spherical gel sample; (**d**) histogram of surface Cu crystal length distribution.

**Figure 9 molecules-27-07261-f009:**
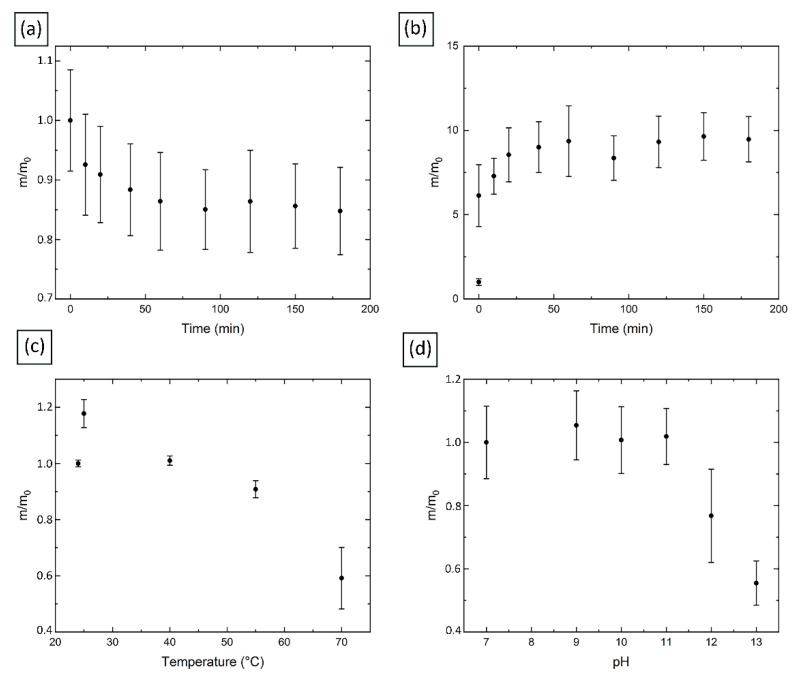
(**a**) Freshly prepared Cu/CS spherical gel sample swelling in water (high molecular weight chitosan, 0.116 M, 100 μL); (**b**) Freeze-dried Cu/CS spherical gel sample swelling in water; (**c**) Effect of temperature on the water swelling capacity of freshly prepared Cu/CS spherical gel sample; (**d**) Effect of pH on the water swelling capacity of freshly prepared Cu/CS spherical gel sample.

**Figure 10 molecules-27-07261-f010:**
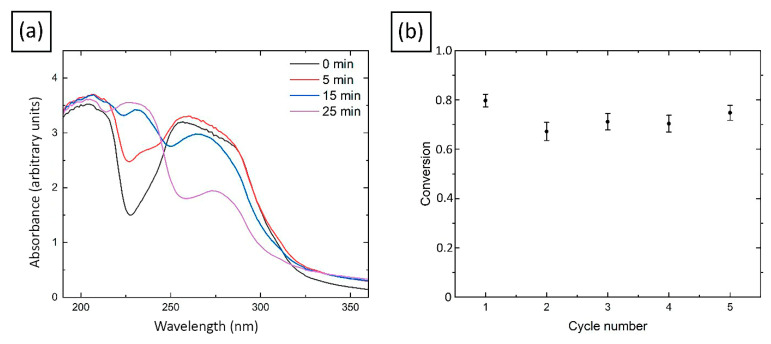
(**a**) UV-vis spectra of nitrobenzene aqueous solution in the presence of NaBH_4_ with Cu/CS catalyst; (**b**) Reusability of Cu/CS catalyst for the nitrobenzene reduction to aniline.

**Table 1 molecules-27-07261-t001:** Antimicrobial activity of Cu/CS composites (Low molecular weight chitosan, molar ratio Cu/CS = 1/10).

Sample	Zone (mm)	
	*B. subtilis* ATCC 6633	*C. albicans* ATCC 2091
Cu/CS	28	17
Amoxiclav/clavulonic acid 20/10 µg	40	-
Amphotericin B 40 µg	-	15

**Table 2 molecules-27-07261-t002:** Densities and porosities of modified Cu/CS lyophilized hydrogels.

Gas	Porosity (%)	Density (g/cm^3^)
H_2_	41 ± 8	0.81 ± 0.05
CO_2_	35 ± 4	0.65 ± 0.04
H_2_ + CO_2_	52 ± 9	0.210 ± 0.012
He + CO_2_	51 ± 6	0.234 ± 0.018

## Data Availability

Not applicable.

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
