# Peer review of "Water Saturated with Pressurized CO2 as a Tool to Create Various 3D Morphologies of Composites Based on Chitosan and Copper Nanoparticles"

_molecules, 2022, doi:10.3390/molecules27217261_

Round 1
Reviewer 1 Report
The article Water saturated with pressurized CO2 as a tool to create various 3D morphologies of composites based on chitosan and copper nanoparticles is suitable for publication in the journal Molecules due to the relevance of developing 3D composites based on organometallic compounds.
The manuscript can be accepted for publication in the journal Molecules after the following comments have been minor revision:
- In section 3.6. Antimicrobial activity does not describe which bacteria and fungi were used for research and where they were taken from.
- Can the resolution of Figure 4 (a-d) be improved?
- Conclusions should be expanded and contain the main results in quantitative reports.
Reviewer 2 Report
Stamer and co-workers have developed pressurized CO2 as a tool to create 3D morphologies of composites based on chitosan and copper nanoparticles. Composite showed catalytic activity in the reduction of nitrobenzene to aniline. A numbers of homogenous and heterogeneous transition-metal systems have been developed for nitrobenzene reduction (one example: Carbohydrate Polymers, 2017, 161, 187-196, similarly several examples are reported). The authors have contributed to this research field. In this study, they discovered composites based on chitosan and copper nanoparticles which show catalytic activity for the reduction of nitrobenzene to aniline and also composites demonstrate antimicrobial activity against both fungi and bacteria. But, this paper is not appropriate for publication in this journal for the following reasons;
As mentioned in the text, a number of heterogenous catalytic systems for reduction of nitrobenzene to aniline have been developed. Thus the novelty does not meet the bar for publication in Molecules. Unfortunately, the author did not mention any advantages of the present protocol over the previously reported catalytic system. More specialized journals may be appropriate.
Reviewer 3 Report
Major comments:
Figure 2. The shear modulus of hydrogels has to be determined at least in triplicate for each gel type and reported in the study. Additionally, the peculiar trend in mechanical performance - coupled with the trend of the peak at UV-vis (line 245) - over time has to be explained and associated with the formation of nanoparticles and copper clusters. Strain sweep tests could be also performed to investigate the resistance to deformation. In this paper, the rheological characterization of the system produced with the high molecular weight chitosan could be also added.
Line 409. A range in the density of scaffolds and films has to be reported. Indeed, it is possible to fabricate structures with very different compositions.
Line 697. The test types and the parameters used to perform tests have to be specified.
Line 733. Check the apex of the number (107). How did you determine the bacteria concentration?
Line 735. Does the agar contain also media for bacteria? Please specify it.
Round 2
Reviewer 2 Report
Firstly, synthesis of 3D morphologies of composites based on chitosan and copper nanoparticles seems to be useful, I don't have deep specialist knowledge of this filed, so hopefully another referee can judge the true novelty and utility in this area.
Specific comments: Authors included few sentences in the manuscript asserting the novelty of their work, so I am happy to recommend publication.
Reviewer 3 Report
The authors improved the overall quality of the present manuscript. Furthermore, the authors addressed the concerns of my previous revision.